# A Starch- and Sucrose-Reduced Diet in Irritable Bowel Syndrome Leads to Lower Circulating Levels of PAI-1 and Visfatin: A Randomized Controlled Study

**DOI:** 10.3390/nu14091688

**Published:** 2022-04-19

**Authors:** Bodil Roth, Julia Myllyvainio, Mauro D’Amato, Ewa Larsson, Bodil Ohlsson

**Affiliations:** 1Department of Internal Medicine, Skåne University Hospital, SE-20502 Malmö, Sweden; bodil.roth@med.lu.se; 2Department of Clinical Sciences, Lund University, SE-22100 Lund, Sweden; julia.myllyvainio@hotmail.se (J.M.); ewa-larsson@outlook.com (E.L.); 3Gastrointestinal Genetics Lab, CIC bioGUNE—BRTA, 48160 Derio, Spain; mdamato@cicbiogune.es or; 4Ikerbasque, Basque Foundation for Science, 48080 Bilbao, Spain; 5Department of Medicine and Surgery, LUM University, 70010 Casamassima, Italy

**Keywords:** AXIN1, cholecystokinin (CCK), enkephalins, ghrelin, irritable bowel syndrome (IBS), plasminogen activator inhibitor-1 (PAI-1), starch- and sucrose-reduced diet (SSRD), visfatin

## Abstract

Irritable bowel syndrome (IBS) is characterized by gastrointestinal symptoms. Overweight and increased risk of metabolic syndromes/diabetes are observed in IBS, conditions associated with plasminogen activator inhibitor-1 (PAI-1) and visfatin. The aim of this study was to measure blood levels of AXIN1, cholecystokinin (CCK), enkephalin, ghrelin, neuropeptide Y (NPY), PAI-1, and visfatin before and after a 4-week intervention with a starch- and sucrose-reduced diet (SSRD). A total of 105 IBS patients were randomized to either SSRD (*n* = 80) or ordinary diet (*n* = 25). Questionnaires were completed, and blood was analyzed for AXIN1 and hormones. AXIN1 (*p* = 0.001) and active ghrelin levels (*p* = 0.025) were lower in IBS than in healthy volunteers at baseline, whereas CCK and enkephalin levels were higher (*p* < 0.001). In the intervention group, total IBS-symptom severity score (IBS-SSS), specific gastrointestinal symptoms, psychological well-being, and the influence of intestinal symptoms on daily life were improved during the study, and weight decreased (*p* < 0.001 for all), whereas only constipation (*p* = 0.045) and bloating (*p* = 0.001) were improved in the control group. PAI-1 levels tended to be decreased in the intervention group (*p* = 0.066), with a difference in the decrease between groups (*p* = 0.022). Visfatin levels were decreased in the intervention group (*p* = 0.007). There were few correlations between hormonal levels and symptoms. Thus, this diet not only improves IBS symptoms but also seems to have a general health-promoting effect.

## 1. Introduction

Irritable bowel syndrome (IBS) is a disorder of a gut-brain interaction (DGBI) characterized by intermittent abdominal pain and altered bowel habits [1], mainly affecting women at a prevalence of 4–11% [2,3]. Dietary adjustment is the primary treatment, although 20–50% of IBS patients still experience gastrointestinal symptoms when following current guidelines [4]. The role of gut and adipose hormones in IBS is not fully understood, and objective measurable signs in IBS patients are few [5,6,7]. However, higher prevalence of the metabolic syndrome, obesity, fibromyalgia, anxiety, depression, increased waist circumference, increased risk of type 2 diabetes, and elevated levels of C-peptide and insulin are found in IBS patients compared to controls [8,9,10,11,12]. Genetic studies suggest functional variants of sucrase-isomaltase (*SI*) coding genes in IBS patients [13]. A starch- and sucrose-reduced diet (SSRD) during 4 weeks in IBS patients resulted in lower intake of carbohydrates but increased intake of fat and protein with reduced gastrointestinal and extraintestinal symptoms, weight, and serum levels of C-peptide, insulin, gastric inhibitory peptide (GIP), and leptin [7,14,15].

Endometriosis affects young women and is accompanied with a high prevalence of gastrointestinal symptoms, making the disease difficult to differ from IBS [16]. The apoptosis-related protein AXIN1 is elevated in endometriosis [17]. Before AXIN1 levels can be used as a biomarker for endometriosis to differ between endometriosis and IBS, AXIN1 levels in IBS patients must be determined.

Cholecystokinin (CCK) is released by endocrine cells in the duodenum and jejunum after protein and fat intake, and regulates gallbladder contraction and gastric emptying [18,19]. Previous studies show increased plasma and tissue levels of CCK in IBS patients compared to healthy controls [20,21].

The opioid receptor agonists leucine-enkephalin and methionine-enkephalin are situated in neurons and endocrine cells in different gastrointestinal regions. These agonists interact with pathways of the enteric nervous system (ENS) that regulate gastrointestinal motility and secretion, whereby gastric emptying is dampened, and the migrating myoelectric complex is disturbed, leading to constipation [22]. Enkephalinase inhibitors are suggested to be an efficient treatment in diarrhea-predominant IBS (IBS-D) [23].

Ghrelin and Neuropeptide Y (NPY) regulate appetite, food intake, and energy metabolism, and are of importance for the development of eating disturbances [24,25]. Ghrelin accelerates gastrointestinal motility and stimulates secretion of gastric acid [24]. Both plasma and tissue levels of ghrelin are found to be elevated in IBS patients [26], whereas another study showed higher density of ghrelin cells in IBS-D and lower in constipation-predominated IBS (IBS-C), compared to controls [27]. 

The adipokines’ plasminogen activator inhibitor-1 (PAI-1) and visfatin are mainly secreted by adipocytes, but also by endothelial cells, leucocytes, hepatocytes, skeletal muscles, and bone marrow [28,29]. Adipokines are involved in the development of obesity, diabetes, inflammation, autoimmunity, and metabolic syndromes [30].

PAI-1 levels are increased among patients with type 2 diabetes in comparison with controls, and form a link between obesity, insulin resistance, and cardiovascular events [29,31].

Our hypothesis was that the changes in food content after SSRD with decreased carbohydrate intake and increased fat and protein intakes [15] should influence the secretion of CCK, enkephalins, ghrelin, NPY, PAI-1, and visfatin, and thereby affect the gastrointestinal symptoms in IBS. 

The primary aim of the present study was to measure blood levels of these hormones before and after a 4-week dietary intervention with SSRD. Secondary aims were to compare the levels of hormones and AXIN1 with healthy volunteers, and to correlate the differences in hormonal levels with changes in gastrointestinal symptoms, psychological well-being, sweet craving, nutritional intake, and weight.

## 2. Materials and Methods

### 2.1. Study Design and Subjects

Patients with IBS (K58.0 and K58.9) diagnosed by their ordinary clinician were identified from registries of primary healthcare centers (PCC) and the Department of Gastroenterology, Skåne University Hospital, Malmö. In total, 2034 IBS patients from PCC were identified. After exclusion of duplicates, 1039 patients remained. Invitation letters were sent to 528 patients after exclusion of all patients with names suggesting an ethnicity outside Scandinavia/Northern Europe, patients living outside the closest neighborhood of the cities Lund and Malmö, or patients whose telephone numbers could not be found. From the tertiary healthcare center, 789 patients were identified. After exclusion of duplicates, 640 patients remained. Invitation letters were sent to 151 patients according to the same criteria as stated above. Patients were contacted by mail and telephone (Figure 1).

Patients who were willing to participate (*n* = 145; 112 patients (77.2%) from PCC; 34 men (23.4%)) were sent a package of study questionnaires to complete prior to an appointment at the Internal Medicine Research Group, Skåne University Hospital, Malmö. After further exclusions (Figure 1), a total of 105 IBS patients (23 men (21.9%)) who fulfilled the inclusion criteria (77 patients (73.3%) from the PCC) were finally enrolled in the study from the 679 invitation letters sent (15.5% inclusion rate) and randomized to either the intervention (*n* = 80) or the control group (*n* = 25) (Figure 2). Of these, 97 participants completed the study (Figure 2)

Patients in the intervention group followed a SSRD for 4 weeks. Verbal and written dietary advice were provided at the start of the study. The dietary advice was modified from guidelines for patients with congenital sucrase-isomaltase deficiency (CSID) [32]. Controls were instructed to maintain their ordinary eating habits, i.e., frequency and regularity in intake and type of food. Blood samples were collected during non-fasting conditions at baseline and at the end of the study, collected at the same time-point for each participant on both occasions, for analyses of AXIN1, CCK, visfatin, enkephalin, ghrelin, NPY, PAI-1, and visfatin, along with multiple questionnaires and 4-day food registrations (from baseline, day 7–10, and day 25–28) (Figure 1).

### 2.2. Dietary Advice

The dietary advice given to the randomized intervention group (*n* = 80) primarily focused on starch and sucrose reduction, with decreased intake of foods such as confectionaries, sweetened dairy products, processed/ultraprocessed foods, bread, pasta, and rice. Instead, participants were advised to continue with or increase their intake of all meats and fish, fat, natural dairy products, eggs, nuts and seeds, and selected berries, fruits, and vegetables low in starch and sucrose (Table 1) [32]. Gluten and lactose contents in the ingested products were unrestricted. Fiber-rich bread, raw rice, and fiber-rich pasta were preferred instead of white bread and ordinary rice and pasta to delay the nutrient transport through the gastrointestinal tract. Adding fat and/or protein to starch-rich foods was also recommended to enhance starch tolerance by delayed gastrointestinal transport, and thus, longer exposure time to digestive intestinal enzyme activity in the small intestine. In general, participants were recommended to restrict their intake of fiber-rich cereals to a maximum of one serving per day. The patients were encouraged to eat slowly and chew their food properly to increase the secretion of amylase, which can contribute to degradation of starch. All participants, both the intervention and control groups, were encouraged to maintain their physical activity level according to their habits before the study. They were also encouraged to continue with their ordinary medications.

### 2.3. Questionnaires

A study questionnaire comprising sociodemographic factors, family history, lifestyle habits, dietary habits, medical health, and medical treatments was completed at baseline.

The Rome IV questionnaire (questions No 40–48 in the Swedish version) was utilized to diagnose DGBI, and license to use it was approved by the Rome Foundation, Inc., Raleigh, NC, USA [33].

The participants registered all types and amounts of food and liquid ingestion and time point of intake during 4 days prior to study start and during 4 days at the termination of the intervention (Figure 1). Experienced gastrointestinal symptoms in connection with their food intake were noted. Information was provided regarding the percentage of fat in dairy products, fiber in bread products, and cacao in chocolate. Type of soda (sugar-free or regular) consumed was noted, and the product manufacturer was given when applicable. For each patient, nutrient intake was calculated from a single day (day 2) of the 4-day registrations. Daily nutrient intake calculations, in total amounts of grams and energy percentages (E%), were performed by a nutritionist, using the AIVO Diet computer program from the National Food Agency, Sweden [34].

The irritable bowel syndrome-symptom severity score (IBS-SSS) comprises questions regarding abdominal pain, abdominal distension, satisfaction with bowel habit, and the impact of bowel habits on daily life, answered on visual analogue scales (VAS), where scores close to 0 mm suggest “no symptoms”, and scores close to 100 mm suggest “severe symptoms”. In addition, there is a question about the number of days with abdominal pain in the last 10 days. The maximum score is 500. Scores 75–174 suggest a slight disease; scores 175–299 suggest a modest disease; and scores ≥300 suggest a severe disease [35].

The visual analog scale for irritable bowel syndrome (VAS-IBS) is a validated questionnaire comprising abdominal pain, diarrhea, constipation, bloating and flatulence, vomiting and nausea, intestinal symptom’s influence on daily life, and psychological well-being on scales of 0–100 mm, where 0 mm represents no symptoms, and 100 mm represent maximum severity. The scales are inverted from the original version [36].

Sweet craving was estimated by a VAS scale where 0 mm means no sweet craving and 100 mm maximal craving [37].

### 2.4. Healthy Volunteers

Healthy volunteers were recruited from the same region, and they had to complete the study questionnaire and VAS-IBS to assure that they were healthy. Forty-eight healthy volunteers served as controls for the AXIN1 values, 26 women (54.2%), age 43.9 ± 13.3 years, and weight 80.5 ± 18.9 kg. The prevalence of women was lower in the healthy volunteers compared with the patients (*p* = 0.003), whereas age and weight did not differ between healthy volunteers and patients (*p* = 0.361 and *p* = 0.075, respectively). Sixty-six healthy volunteers served as controls for the hormonal analyses, 54 women (81.8%), age 39.6 ± 11.9 years, and weight 67.7 ± 12.5 kg. There was no difference in sex distribution between healthy volunteers and IBS patients (*p* = 0.697). Age and weight were lower in healthy volunteers than in patients (*p* = 0.003 and *p* = 0.014, respectively).

### 2.5. Hormonal Analyses

Blood samples were taken prior to and after the dietary intervention in IBS patients, and at one time point in healthy volunteers, and serum and EDTA plasma were stored at −20 °C or −80 °C until analysis. Serum levels of CCK and PAI-1, and EDTA plasma levels of enkephalin, were measured using enzyme-linked immunosorbent assay (ELISA), and EDTA plasma ghrelin and serum NPY and visfatin were measured using Meso Scale Discovery (MSD) (mesoscale) (Table 2). Intra-assay and inter-assay coefficients of variance (CV) are shown in Table 3. As there were no suitable samples for calculating CV on ghrelin, this is missing.

A complete description of the analyses performed according to the manufacturer’s instructions is given in Appendix A. Briefly, measurement of serum CCK was carried out by ELISA (serial number 2D4E070E57, Cloud-Clone Corp., Oxfordshire, UK). Standard or sample and detection reagent A were added to the plate. After a washing procedure, detection reagent B was added.

To measure enkephalin by ELISA (Cusabio, Fannin, Houston, TX, USA), the standard or sample was added to each well. After the incubation, the liquid was removed from the wells. Biotin-antibody was added, and after incubation and washing, horseradish (HRP)-avidin was added. The washing process was repeated, and TMB substrate was added.

PAI-1 was measured by ELISA (Thermo Fisher Scientific, Waltham, MA, USA). Assay buffer was added to the standard wells and the blank wells. Thereafter, the prepared standard was added to the first standard well, which was mixed and transferred to each standard well. The assay buffer and prediluted sample were added to each sample well. The plate was incubated with Biotin-conjugate. After washing, streptavidin-HRP was added. The washing process was repeated, and TMB substrate was added.

After incubations in the ELISAs, the reaction was stopped, and measurements were immediately conducted at 450 nm optical density by a microplate reader.

The Mesoscale Discovery (MSD, Rockville, MD, USA) R-PLEX singleplex or multiplex assay and U-PLEX singleplex assay metabolic group (Human) were used to perform the analyses of visfatin, NPY, and ghrelin (total and active), respectively, by electro-chemiluminescence detection [38]. The intensity of emitted light is proportional to the amount of ghrelin, NPY, or visfatin in the wells.

The biotinylated ghrelin capture antibody was added to the MSD GOLD small spot streptavidin plate and incubated overnight. A metabolic assay working solution (MWS) containing aprotinin, DPP-IV inhibitor (DPP4, Merck, Darmstadt, Germany), and diluent for the dilution of the calibrator and samples of ghrelin active and aprotinin in diluent for ghrelin total was prepared for dilution of the calibrator and EDTA plasma sample. The calibrator (active 9070–2.2 pg/mL, total 5180–1.3 pg/mL) and EDTA plasma were added after the plates were washed. Incubation in RT was followed by a new washing procedure, and a SULFO-TAG detection antibody was added. After incubation and a washing procedure, 150 μL MSD GOLD read buffer B in each well were added, and the plates were read on a MSD instrument.

The biotinylated NPY capture antibody was coupled with a linker added to each well on a U-PLEX plate and incubated overnight. The calibrator (500,000–122 pg/mL) and serum sample were added after the plates were washed three times with MSD wash buffer. Incubation in RT was followed by a new washing procedure, and a SULFO-TAG detection antibody was added. After a new incubation and a washing procedure, 150 μL MSD GOLD read buffer A in each well were added, and the plates were read on a MSD instrument.

The biotinylated visfatin capture antibody was added to the MSD GOLD small spot streptavidin plate and incubated overnight. The calibrator (2500–0.61 ng/mL) and serum sample were added after the plates were washed with MSD wash buffer. Incubation in RT was followed by a new washing procedure, and a SULFO-TAG detection antibody was added. After incubation and a washing procedure, 150 μL MSD GOLD read buffer in each well were added, and the plates were read on a MSD instrument.

### 2.6. AXIN1 Analysis

Plasma AXIN1 was analyzed by a separate ELISA (MBS762601, MyBiosource, San Diego, CA, USA) as described in detail previously [17].

### 2.7. Statistical Analyses

All calculations were carried out using SPSS (version 25; IBM Corporation). Normality was calculated by the Kolmogorov–Smirnov test. Age and weight were normally distributed and calculated by Student’s *t*-test or paired-samples *t*-test. Other continuous parameters were calculated by the Mann–Whitney U-test, Wilcoxon signed ranks test, or Spearman´s correlation test, and Fischer´s exact test was used for binary variables. The generalized linear model was used to calculate differences between healthy volunteers and patients (predictor) at baseline, adjusted for sex regarding AXIN1 and age and weight regarding hormonal analyses (dependent variables). Values are given as mean ± standard deviation (SD), median and interquartile range (IQR), or β-values and a 95% confidence interval (CI). The *p*-values < 0.05 were considered statistically significant.

## 3. Results

### 3.1. Basic Characteristics

Of the 105 included patients, 86 fulfilled the Rome IV criteria for IBS [1]. Thirty-seven patients had mixed IBS (IBS-M); 26 patients had IBC-D; 20 had IBS-C; and three patients had unspecified IBS (IBS-U). Seventeen patients had DGBI (i.e., two or more of the following were not present at least 30% of the time: pain associated with improvement or worsening with defecation, changed consistency of stool, or changed frequency of defecation), and two patients did not complete the Rome IV questionnaire. The duration of IBS was 17.5 (9.0–28.0) years. In addition to IBS, the patients suffered from allergy (*n* = 17), asthma bronchialis (*n* = 11), depression (*n* = 11), hypertension (*n* = 10), migraine (*n* = 7), lactose intolerance (*n* = 5), muscle pain (*n* = 5), and type 2 diabetes (*n* = 1). The patients used antidepressants (*n* = 21), laxatives (*n* = 14), proton pump inhibitors (*n* = 13), and levothyroxine (*n* = 13). One patient suffered from endometriosis, and three used birth control medications.

More women than men were enrolled, but the age and sex distribution did not differ between groups (Table 4). In the intervention group, 40 patients (51.3%) exercised at most 60 min/week; 20 patients (25.7%) exercised 60 to 120 min/week; and 18 patients (23.1%) exercised >120 min/week. In the control group, the corresponding figures were 11 patients (44.0%), 6 patients (24.0%), and 8 patients (32.0%), respectively (*p* = 0.838). The weight was slightly lower in the control group compared with the intervention group at baseline. The weight was decreased in the intervention group during the study, which was not found in the control group (Table 4).

### 3.2. Dietary Intake

The dietary intake was equal at baseline, except for a slightly lower protein intake in the control group. After 4 weeks, the carbohydrate, starch, and sucrose ingestions were significantly lowered, and fat and protein intakes were increased, in the intervention group, whereas no differences were found in the control group. The intakes of carbohydrates, starch, and sucrose differed between the groups at week 4 (Table 4). 

### 3.3. Gastrointestinal Symptoms and Sweet Craving

In the intervention group, 73.8% were responders, as defined by a decrease in total IBS-SSS of ≥ 50 points, and 18.8% had no symptoms with <75 in total IBS-SSS after 4 weeks, according to the classifications [35]. In the control group, 24.0% were responders, and none were out of symptoms after 4 weeks (*p* < 0.001). The degree of gastrointestinal symptoms was equal at baseline. During the intervention, all the symptoms were decreased, and the psychological well-being was improved, in the intervention group, whereas only constipation and bloating and flatulence were improved in the control group. Significant differences between the groups after 4 weeks were found in total IBS-SSS, abdominal pain, bloating and flatulence, vomiting and nausea, and the intestinal symptoms’ influence on daily life (Table 4). The sweet craving in the control group was 51 (34–70) mm at baseline and 57 (30–70) mm after 4 weeks (*p* = 0.671). In the intervention group, the sweet craving was 60 (32–79) mm at baseline and 21 (10–42) mm after 4 weeks (*p* < 0.001). The 4-week values differed between the groups (*p* < 0.001), as did the changes between the groups (Figure 3).

### 3.4. AXIN1 Levels

The AXIN1 values were higher in men than in women (199.0 (90.2–261.0) pg/mL vs. 116.4 (72.2–225) pg/mL; *p* = 0.026). AXIN1 levels were significantly lower in IBS compared to healthy volunteers (96.9 (64.8–158.9) pg/mL vs. 219.0 (176.0–281.2) pg/mL; *p* = 0.001). This difference remained after adjustment for sex (Figure 4, Table 5), and the AXIN1 values were higher in healthy volunteers even when only women were included in the analysis (*p* = 0.005). The AXIN1 levels were higher in the control group than the intervention group and unaffected by the dietary intervention (Table 6). There was an inverse correlation between AXIN1 and disease duration (*rs* = (−0.228), *p* = 0.043), but no correlations between AXIN1 and age, weight, nutrition intake, gastrointestinal symptoms, or hormonal factors at any time point or in differences between baseline and week 4 could be found. The AXIN1 levels did not differ between those with or without hormonal treatment or between subgroups of IBS (data not shown).

### 3.5. Hormonal Levels at Baseline

CCK and enkephalin levels were higher in IBS patients than in healthy volunteers, whereas active ghrelin was slightly lower in IBS (Figure 4, Table 5). NPY levels were too low to be detectable (data not shown).

Within the IBS patients at baseline, age correlated inversely with visfatin (*rs* = (−0.216), *p* = 0.038) and weight inversely with visfatin (*rs* = (−0.228), *p* = 0.032) and total ghrelin (*rs* = (−0.310, *p* = 0.003), whereas age correlated positively with enkephalin (*rs* = 255, *p* = 0.013), and weight correlated positively with PAI-1 (*rs* = 0.348, *p* = 0.001). There was a correlation between CCK and constipation (*rs* = 0.219, *p* = 0.036) and between total ghrelin and total IBS-SSS (*rs* = 0.231, *p* = 0.026) and constipation (*rs* = 0.230, *p* = 0.026). No correlations were observed for hormonal levels and dietary intake (data not shown).

### 3.6. Hormonal Levels during the SSRD Intervention

CCK levels were higher in the intervention group compared to the control group of the IBS cohort. During the study, the CCK levels were increased in the control group but unaffected in the intervention group (Table 6).

Enkephalin and ghrelin levels were unaffected during the study (Table 6). There was a tendency toward decreased PAI-1 levels in the intervention group, and the change in hormonal levels of PAI-1 between baseline and week 4 differed between the intervention and control group (*p* = 0.022) (Figure 5, Table 6). Visfatin levels were markedly reduced in the intervention group, without any changes in the control group (Table 6).

Regarding correlations of changes during the study (delta value), there was a positive correlation between the change in weight and CCK (*rs* = 0.241, *p* = 0.026), and a negative correlation between the change in fat intake and enkephalin (*rs* = (−0.275), *p* = 0.008). When further studying the fat composition, enkephalin correlated negatively with both saturated fat (*rs* = (−0.260, *p* = 0.012) and mono-unsaturated fat (*rs* = (−0.211), *p* = 0.044). The sweet craving was inversely correlated with visfatin levels (*rs* = (−0.327), *p*= 0.002) and positively associated with PAI-1 levels (*rs* = 0.267, *p* = 0.012). The only correlation with reduced symptoms was an inverse correlation between constipation and ghrelin (*rs* = (−0.217), *p* = 0.037). All other correlations between changes in hormonal levels and changes in symptoms, nutrient intakes, or weight were nonsignificant (data no shown).

## 4. Discussion

The main findings of the present study were that IBS patients had higher levels of CCK and enkephalin and lower levels of AXIN1 and active ghrelin compared with healthy volunteers. During the SSRD intervention, levels of PAI-1 and visfatin were lowered along with improvements in gastrointestinal symptoms and psychological well-being and lower weight and sweet craving.

IBS is not only characterized by gastrointestinal symptoms. A large cross-sectional study reported that IBS patients had a significantly higher prevalence of the metabolic syndrome, overweight, and obesity, and increased waist circumference in comparison with control subjects [11]. Further, IBS patients had increased risk of type 2 diabetes and elevated levels of C-peptide and insulin compared to controls [8,9,10].

The current reduction in PAI-1 levels is interesting, since animal studies suggest PAI-1 to play a causal role in the development of insulin resistance and the metabolic syndrome [39]. A systemic review summarized that PAI-1 levels were elevated in type 2 diabetes and could be a potentially significant but underestimated risk factor for diabetes [29]. In a longitudinal population-based study, elevated PAI-1 levels preceded diabetes [31]. Furthermore, lifestyle- and dietary-mediated weight loss in overweight and moderately obese subjects have in several studies been associated with reductions in PAI-1 levels [29].

Visfatin stimulates insulin secretion and increases insulin sensitivity. Thereby, visfatin stimulates glucose uptake by muscle cells and adipocytes [28,40,41]. Visfatin is up-regulated in inflammation, and is an important mediator in the production of both pro-inflammatory and anti-inflammatory cytokines [28]. Some studies suggest an increased visfatin concentration in obese subjects, whereas other studies show lowered plasma levels in obesity [42,43].

PAI-1 and visfatin levels in IBS patients have never been examined to our knowledge, and very few studies have examined postprandial visfatin levels [44]. Interrupted balance in adipose tissue is considered to be responsible for the secretion of pro-inflammatory cytokines which are markers of low-grade inflammation and are associated with the development of insulin resistance [28]. The decreased PAI-1 and visfatin levels may thus reflect a diminished low-grade inflammation after SSRD [15]. Local inflammatory differences not detectable in circulating hormone levels may explain the poor correlations between hormone levels and clinical findings but may still result in reduced painful stimuli [12]. When a carbohydrate-reduced diet led to weight reduction in type 2 diabetes, a similar decrease in PAI-1 levels correlated with decreases in glucose and triglyceride levels, whereas a decrease in visfatin levels was found without any correlations with other measured parameters [45].

The decrease in PAI-1 and visfatin in the intervention group may reflect improved metabolic control after replacement of candies and sweetened products with fruits and vegetables along with weight reduction and lower levels of C-peptide, insulin, GIP, and leptin [7,46]. Thus, the SSRD not only improves the gastrointestinal and extra-intestinal symptoms in IBS [14,15], but it also has a general health-promoting effect on the weight, metabolism, and endocrine profile [7]. The greatest changes in hormonal concentrations during the SSRD intervention were found in c-peptide, insulin, GIP, and leptin levels as a response to the great changes in carbohydrate intake [7,15]. All adipokines measured, leptin, PAI-1, and visfatin were reduced after the dietary intervention of reduced starch and sucrose [7], which point to a great importance of the adipose tissue during this intervention. Adjustments of these hormones may serve as a platform for prevention of obesity, the metabolic syndromes, and prediabetes seen in IBS patients [8,9,10,11,12]. In a rodent pre-diabetes model, hyperexcitability was found in the tibial nerve which correlated with weight, insulin resistance, and insulin and leptin levels [47], suggesting that metabolic changes may drive early axonal dysfunction. Altogether, metabolic imbalance and altered homeostasis after a diet high in sweeteners [46] may be of importance for hypersensitivity and the pain report and could possibly explain some of the symptoms in IBS [12,48].

Processed food with added sweeteners and fat have been shown to be the highest risk factor for developing food addiction, with suggestions of cross-sensitization with drugs [49]. Consumption of high-calorie beverages, which was observed to a great extent in our cohort at baseline [46], exerted the same brain response in overweight/obese subjects as addictive drugs [49,50]. Therefore, it is not surprising that sweet craving is associated with overweight/obesity, and that sweet craving was reduced in the current study along with the diminished intake of sweetened and processed food. Both food habits and sickness behavior are socially learned behaviors that affect the development and experience of symptoms from varying organs [49,51]. Thus, improvement in food habits in the society is rudimentary for the health, independent of IBS or not. If not strictly accomplished to the guidelines for CISD, the SSRD recommendations with reduced intake of starch and sucrose are in line with the 2012 Nordic nutrition recommendations published by the National Food Agency in Sweden [52].

CCK levels were higher in IBS patients than in healthy volunteers, in accordance with previous research in both fasting and postprandial plasma and in the sigmoideum tissue [20,21]. The lower CCK levels at baseline in the control group compared with the intervention group may be explained by the lower protein intake in this group [53]. The relatively small increases in fat and protein intake during the study left the CCK levels unaffected in the intervention group. CCK is involved in gastrointestinal motility supporting the correlation between CCK and constipation, and CCK receptor antagonists are under development for treatment of constipation [54].

The increased fat intake in the SSRD trial was mainly dependent on an increase in poly-unsaturated fat intake [55]. Dietary fat modulates brain expression of enkephalin in rats [56,57] as well as the enkephalinase activity [58]. Further, opioid receptors are reported to play an important role in the food reward system in rats, where sweetened products are important [49,50]. No studies of enkephalin and fat metabolism have been performed in humans that can explain the correlation between changes in fat intake and enkephalin levels.

Ghrelin levels were unaffected by the SSRD, which may depend on the fact that all macronutrients contribute to the inhibition of ghrelin secretion, and thus, food *per se* inhibits the secretion, independently of the composition [19].

AXIN1 is a repressor of the Wnt signaling pathway, where it is responsible for several processes such as growth and cell proliferation [59,60]. Dysregulation of the Wnt pathway is found in several diseases such as intestinal inflammation and autoimmune disorders [61,62]. We recently found that AXIN1 is elevated in patients with endometriosis, which is considered an inflammatory disease, and the levels were associated with the degree and duration of gastrointestinal symptoms [17]. The current negative correlation between AXIN1 and the long disease duration of IBS must be compared with the shorter disease duration of endometriosis [17]; the low-grade inflammation in IBS may be more apparent in the early stage of the disease and is too low to affect AXN1 levels [63]. A recent study found elevated AXIN1 levels in inflammatory bowel disease but not in IBS [64]. The present finding together with other studies support the hypothesis that AXIN1 may be a potential biomarker to differ between IBS and endometriosis in young women, two groups of patients with gastrointestinal symptoms but normal levels of fecal calprotectin [17,64]. It was not possible to find any association that could explain the lower AXIN1 levels in IBS patients compared with healthy volunteers. Neither could the difference between the control and intervention group be explained, which could be a Type 1 error, due to the small control cohort.

The strength of the present study is the prospective character of the study along with a dietary intervention. The quality of the analyses was of high quality as found by low CVs, except for the ghrelin analyses, for which the method provided by the manufacturer was more uncertain. One limitation is that the blood samples were not obtained during fasting conditions. However, the adipokines are not that dependent on time point in relation to dietary intake as gut hormones [28]. Furthermore, the patients were met at the same time point at both meetings and compared with themselves. There was an unequal allocation of subjects to intervention and control groups because we wanted as many participants as possible available for analyses of functional *SI* genes in relation to the dietary effect. To adjust for these limitations, calculations were performed not only between the study groups, but also within the groups.

## 5. Conclusions

SSRD improves gastrointestinal symptoms and psychological well-being, at the same time as weight is reduced with an improved endocrine profile, which hypothetically leads to reduced risk of the metabolic syndrome and type 2 diabetes. The current findings stress the importance of replacing starch and sucrose with fruits, vegetables, and dairy products. Metabolic disturbances may be of importance for hypersensitivity and pain reporting [48]. This simple and easy to follow regime [65] improves not only the IBS symptoms but has a general health-promoting effect with improved nutrient intake [15] and improvement in pain syndromes, psychological well-being, and fatigue [14], and weight, metabolism, and endocrine profile [7]. Due to the high prevalence of these diseases in the society, the dietary changes could have a great impact on health and the economic burden in the society and should be considered in the general population as well as in IBS.

## Figures and Tables

**Figure 1 nutrients-14-01688-f001:**
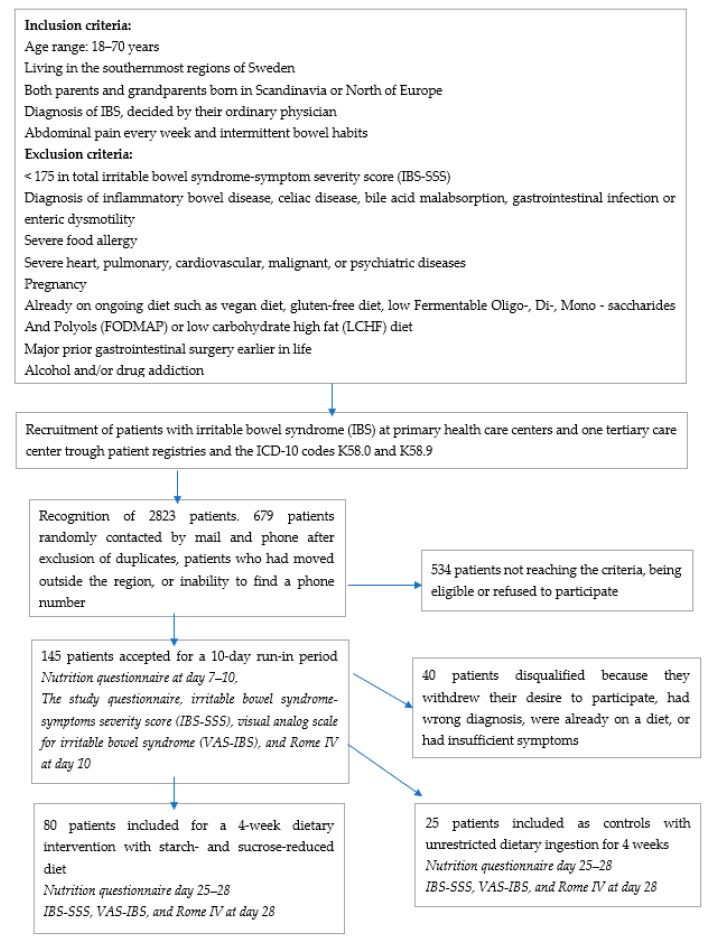
Study design, inclusion and exclusion criteria, and recruitment process of patients with irritable bowel syndrome (IBS) at primary healthcare centers and at one tertiary healthcare center.

**Figure 2 nutrients-14-01688-f002:**
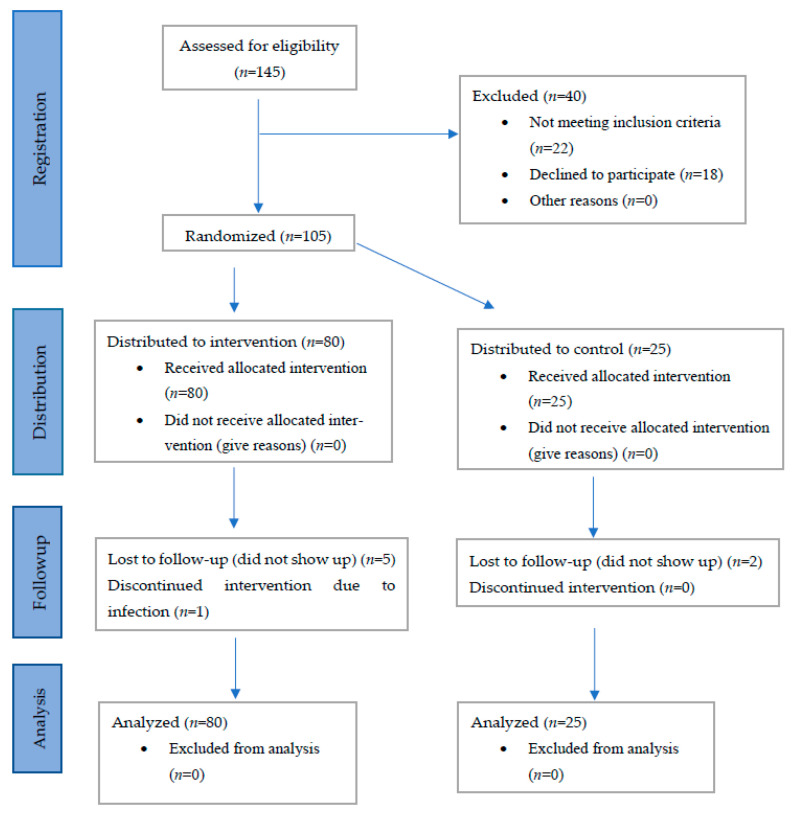
CONSORT flow chart.

**Figure 3 nutrients-14-01688-f003:**
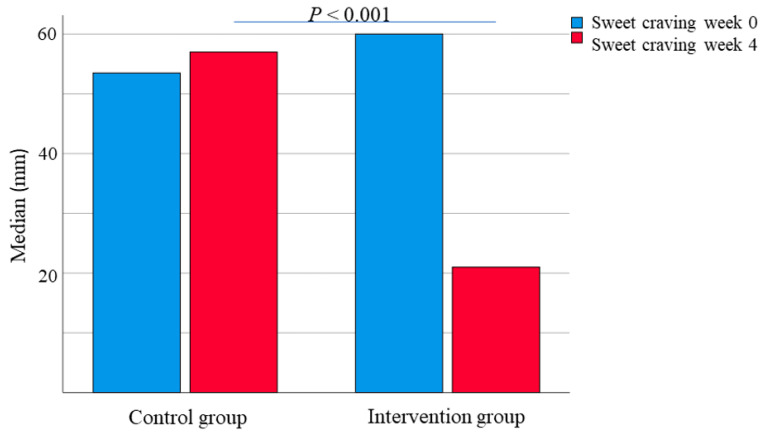
Sweet craving, estimated on a VAS-scale 0–100 mm, in the control and intervention group before and after a 4-week dietary intervention. Wilcoxon signed ranks test. *p* < 0.05 was considered statistically significant.

**Figure 4 nutrients-14-01688-f004:**
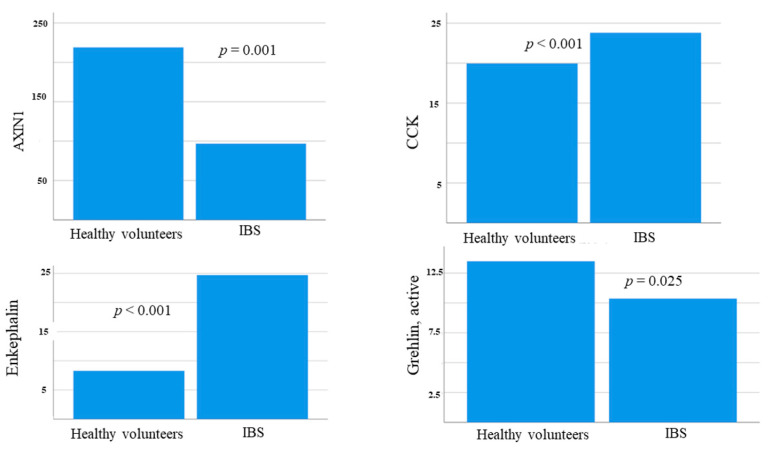
Basal levels (pg/mL) of AXIN1, cholecystokinin (CCK), enkephalin, and active ghrelin in healthy volunteers and IBS patients at baseline. Mann–Whitney U-test. *p* < 0.05 was considered statistically significant.

**Figure 5 nutrients-14-01688-f005:**
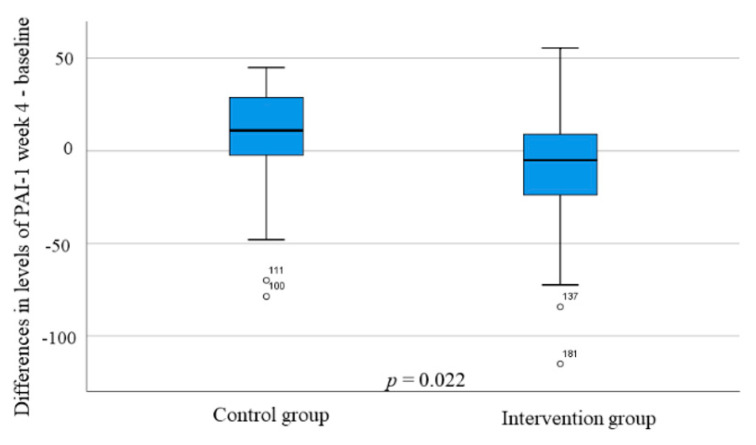
The differences in plasminogen activator inhibitor-1 (PAI-1) (ng/mL) changes during the study between the control and intervention group. Mann–Whitney U-test. *p* < 0.05 was considered statistically significant.

**Table 1 nutrients-14-01688-t001:** Guidelines for berries, fruit, legumes, and vegetable ingestion according to a starch- and sucrose-reduced (SSRD) diet.

**Highly Tolerated**
Berries and fruit: avocado, blackberries, blueberries, boysenberries, cherries, cranberries, currants, figs, gooseberries, grapes, kiwi fruits, lemons, limes, loganberries, olives, papayas, pears, pomegranates, prunes, raspberries, rhubarbs, and strawberries.
Vegetables and legumes: alfalfa sprouts, artichokes*, arugulas, asparagus, bamboo shoots, bok choy, broccoli *, brussel sprouts *, cabbages *, cauliflower *, celery, chard, chicories, chives, collard greens, cress, cucumbers, eggplants, endive, green beans, kale, lettuces, mung bean sprout, mushrooms, peppers, radishes, spaghetti squash, spinach, tomatoes, turnips, yellow squash, zucchini.
**Tolerated by a Few**
Berries and fruit: persimmons, plums, raisins, watermelon.
Vegetables and legumes: edamame soybeans, jicamas, leeks, okra, pumpkins, snow peas, tempeh, tofu, yellow wax beans.
**Not Tolerated**
Berries and fruit: apples, apricots, bananas, cantaloupe, dates, grapefruits, guava, honeydew melon, mangos, nectarines, oranges, passion fruits, peaches, pineapples, tangelos, tangerines
Vegetables and legumes: beets, black beans, black-eyed peas, butternut, carrots, cassavas, chickpeas, corn, garlic, green peas, lentils, kidney beans, lima beans, navy beans, onion, parsnips, pinto beans, potatoes, soybeans, split peas, sweet potatoes, yams.

Information from Ref No 32. Bold headings refer to the division of all fruits, berries and vegetables into three groups, depending on tolerability. * = vegetables which might result in gas production in all subjects.

**Table 2 nutrients-14-01688-t002:** Description of antibodies.

Analysis Description	Catalog Number of Antibody Set Used	Detection Range
hu CCK	CEA802Hu	12.35–1000 pg/mL
hu Ghrelin, total	MSD U-PLEX	1.3–5180 pg/mL
hu Ghrelin, active	MSD U-PLEX	2.2–9070 pg/mL
hu NPY	MSD R-PLEX	122–500,000 pg/mL
hu PAI-1	BMS2033 and BMS2033TEN	78–5000 pg/mL
hu Proenkephalin	CSB-EL017781HU	15.6–1000 pg/mL
hu Visfatin	MSD R-PLEX	0.6–2500 ng/mL

Hu = human, CCK = cholecystokinin, NPY = neuropeptide Y, PAI-1 = plasminogen activator inhibitor 1.

**Table 3 nutrients-14-01688-t003:** Intra-assay and inter-assay coefficients of variance (CV) from immunological analyses.

	Intra-Assay CV	Inter-Assay CV
CCK CV (%)	*n* = 8 <10	*n* = 24 <12
PAI-1 CV (%)	*n* = 6 4.7	*n* = 18 5.0
Proenkephalin CV (%)	*n* = 20 <8	*n* = 20 <10
Visfatin CV (%)	*n* = 8 4.5	*n* = 5 14.4

*n* = number, CCK = cholecystokinin, PAI-1= plasminogen activator inhibitor 1.

**Table 4 nutrients-14-01688-t004:** Basic characteristics.

Characteristics	Control Group (*n* = 25)	*p* Value *	Intervention Group (*n* = 80)	*p* Value *	*p* Value **
**Sex** (women/men; *n*, %)	22 (88)/3 (12)		60 (75)/20 (25)		0.267
**Age** (years)	41.4 ± 14.5		47.5 ± 12.4		0.064
**Carbohydrate intake** (g)					
Baseline	177 (112–207)		185 (144–223)		0.402
Week 4	182 (89–224)	0.235	88 (66–128)	<0.001	<0.001
**Fat intake** (g)					
Baseline	61 (46–72)		65 (44–94)		0.394
Week 4	69 (46–96)	0.566	72 (56–104)	0.049	0.264
**Protein intake** (g)					
Baseline	59 (46–71)		72 (55–83)		0.046
Week 4	65 (53–81)	0.571	82 (58–99)	0.023	0.127
**Sucrose intake** (g)					
Baseline	20 (13–43)		23 (13–38)		0.823
Week 4	19 (5–36)	0.144	5 (2–13)	<0.001	<0.001
**Starch intake** (g)					
Baseline	71 (43–90)		76 (49–116)		0.448
Week 4	82 (37–101)	0.849	22 (3–48)	<0.001	<0.001
**Weight** (kg)					
Baseline	68.3 ± 14.8		75.8 ± 14.9		0.037
Week 4	70.0 ± 14.7	0.141	73.7 ± 14.8	<0.001	0.304
**Total IBS-SSS**					
Baseline	310 (247–351)		306 (250–356)		0.820
Week 4	300 (233–331)	0.248	156 (88–250)	<0.001	<0.001
**Abdominal pain** (mm)					
Baseline	49 (27–63)		52 (37–65)		0.441
Week 4	50 (32–63)	0.650	24 (6–43)	<0.001	<0.001
**Diarrhea** (mm)					
Baseline	47 (5–70)		57 (18–76)		0.312
Week 4	24 (1–49)	0.300	14 (1–33)	<0.001	0.269
**Constipation** (mm)					
Baseline	54 (30–69)		47 (1–73)		0.763
Week 4	28 (1–68)	0.045	18 (0–36)	<0.001	0.146
**Bloating and flatulence** (mm)					
Baseline	78 (68–89)		78 (60–85)		0.429
Week 4	69 (56–80)	0.001	28 (11–54)	<0.001	<0.001
**Vomiting and nausea** (mm)					
Baseline	29 (6–50)		12 (1–37)		0.134
Week 4	12 (2–56)	0.112	3 (0–24)	0.002	0.043
**Psychological well-being** (mm)					
Baseline	47 (24–71)		50 (24–69)		0.950
Week 4	48 (32–60)	0.732	36 (13–53)	<0.001	0.092
**Influence on daily life** (mm)					
Baseline			72 (52–86)		0.539
Week 4		0.732	35 (20–66)	<0.001	<0.001

*n* = number. Two missing values at baseline and six at 4 weeks in the intervention group. Three missing values at 4 weeks for nutrition intake and two for symptoms in the control group. Gastrointestinal symptoms assessed by irritable bowel syndrome-symptoms severity score (IBS-SSS) [35] and visual analog scale for IBS (VAS-IBS) [36]. Variables in bold. The values are presented as number (percentage), mean ± standard deviation, or median and interquartile range. * = Wilcoxon signed ranks test or paired-samples *t*-test (weight), ** = Fisher´s exact test, Mann–Whitney U-test, or Student´s *t*-test. *p* < 0.05 was considered statistically significant.

**Table 5 nutrients-14-01688-t005:** AXIN1 and hormone levels in IBS patients and healthy volunteers at baseline.

	Healthy Volunteers	IBS Patients (*n* = 105)	β	95% CI	*p*-Value
**P-AXIN1 (pg/mL)**	219.0 (176.0–281.2)	96.9 (64.8–158.9)	−117.477	−189.484–(−45.470)	0.001
**S-CCK (pg/mL)** Missing value: 13 in patients	20.0 (17.7–23.0)	23.8 (20.4–28.0)	4.621	2.368–6.874	<0.001
**P-Enkephalin (pg/mL)** Missing value: 21 and 11	8.3 (5.3–11.7)	24.7 (18.3–32.3)	15.410	9.131–21.689	<0.001
**P-Ghrelin active (pg/mL)** Missing value: 23 and 12	13.5 (6.8–24.4)	10.4 (3.4–19.2)	−6.036	−11.307–(−0.765)	0.025
**P-Ghrelin total** **(pg/mL)** Missing value: 33 and 56	408.5 (275.5–789.0)	385.0 (240.5–701.5)	−109.548	−307.153–88.056	0.277
**S-PAI-1 (ng/mL)** Missing value: 2 and 13	124.0 (105.2–156.8)	113.8 (84.3–142.5)	−13.990	−46.356–18.377	0.397
**S-Visfatin (ng/mL)** Missing value: 29 and 12	13.5 (5.2–55.5)	14.6 (5.0–107.4)	758,732	−810,652–2,328,117	0.343

CCK = cholecystokinin, IBS = irritable bowel syndrome, *n* = number, PAI-1 = plasminogen activator inhibitor 1, P = plasma, S = serum. Forty-eight healthy volunteers served as controls for AXIN1 and 66 as controls for the hormone analyses. Missing values represent healthy volunteers and IBS patients, respectively. Generalized linear model was used to calculate differences between healthy volunteers and patients at baseline, adjusted for sex regarding AXIN1 and age and weight regarding the hormonal analyses. Dependent variables in bold. Values are given as median and interquartile range (IQR) and β-values and 95% confidence interval (CI). *p* < 0.05 was considered statistically significant.

**Table 6 nutrients-14-01688-t006:** Levels of AXIN1 and hormones prior and after a 4-week dietary intervention with a starch- and sucrose-reduced diet (SSRD).

Hormones	Control Group (*n* = 25)	*p* Value *	Intervention Group (*n* = 80)	*p* Value *	*p* Value **
**P-AXIN1 (pg/mL)**					
Baseline	142.1 (99.4–292.6)		81.1 (56.7–134.0)		<0.001
Week 4	124.4 (96.4–321.1)	0.899	88.4 (60.8–149.0)	0.491	0.006
**S-CCK (pg/mL)**					
Baseline	21.6 (17.8–26.3)		24.0 (21.6–28.3)		0.027
Week 4	26.6 (18.1–28.0)	0.031	25.0 (21.9–29.5)	0.275	0.209
**P-Enkephalin** **(pg/mL)**					
Baseline	26.4 (18.3–35.3)		24.6 (17.9–31.8)		0.420
Week 4	24.9 (19.4–30.7)	0.482	26.5 (16.2–32.8)	0.999	0.984
**P-Ghrelin, active (pg/mL)**					
Baseline	6.9 (3.5–12.4)		12.7 (3.89–22.0)		0.107
Week 4	6.8 (3.7–14.9)	0.266	9.7 (3.5–22.6)	0.944	0.456
**P-Ghrelin, total** **(pg/mL)**					
Baseline	462.0 (270.0–804.5)		369.0 (223.5–682.0)		0.382
Week 4	484.0 (221.5–805.0)	0.664	393.0 (250.5–665.8)	0.171	0.907
**S-PAI-1 (ng/mL)**					
Baseline	116.6 (67.0–161.6)		113.8 (88.4–141.8)		0.822
Week 4	133.0 (106.8–160.8)	0.129	107.9 (83.1–134.0)	0.066	0.113
**S-Visfatin (ng/mL)**					
Baseline	20.2 (7.8–104.5)		12.0 (4.6–117.6)		0.382
Week 4	19.2 (8.8–105.0)	0.273	10.3 (5.2–114.1)	0.007	0.327

CCK = cholecystokinin, PAI-1 = plasminogen activator inhibitor-1, P = plasma, S = serum. Variables in bold. Values are presented as median and interquartile range. * =Wilcoxon test ** = Mann–Whitney U-test. *p* < 0.05 was dealt with as statistically significant values.

## Data Availability

The data presented in this study are available on request from the corresponding author. The data are not publicly available due to data protection regulation law.

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
