# Peer review of "A Starch- and Sucrose-Reduced Diet in Irritable Bowel Syndrome Leads to Lower Circulating Levels of PAI-1 and Visfatin: A Randomized Controlled Study"

_nutrients, 2022, doi:10.3390/nu14091688_

Round 1

Reviewer 1 Report

In the present article by Roth et al. with the title "A starch- and sucrose-reduced diet in irritable bowel syndrome leads to lower circulating levels of PAI-1 and visfatin: a randomized controlled study", the authors provide a thorough statistical analysis of the hormonal levels in blood and gastrointestinal symptoms of IBS patients before and after dietary intervention with SSRD. While the paper would definitely deserve to be considered for publication in the Special Issue "Gluten-Free Diet and Gastrointestinal Diseases", several issues need to be addressed to finalize it.

The Abstract ends too abruptly. I would recommend the authors to add one more sentence as a conclusion.

In Figures 1 and 2, some parts of the text do not fit the rectangles so they cannot be seen.

The diagrams in Figures 3, 4, and 5 are too small, especially labels on the OY axis. Also, units of measurement should be added.

Table 3 lacks the data on ghrelin, while table 5 lacks the list of abbreviations as in other tables.

Author Response

Dear Reviewer.

Thank you very much for reviewing our manuscript: Manuscript ID: nutrients-1636282;
Title: A starch- and sucrose-reduced diet in irritable bowel syndrome leads
to lower circulating levels of PAI-1 and visfatin: a randomized controlled
study

We have now tried to answer all the questions and think that the manuscript has been improved after revision. All changes are marked in yellow. We have also performed a language correction, but all these small changes throughout the manuscript are not marked. A detailed method description of ELISA and mesoscale is added as supplementary material, since one of the reviewers wanted a shorter method description within the manuscript. We now hope that you will further consider the manuscript for publication in Nutrients.

Bodil Ohlsson, professor, senior consultant in Gastroenterology and Hepatology

Reviewer 1.

In the present article by Roth et al. with the title "A starch- and sucrose-reduced diet in irritable bowel syndrome leads to lower circulating levels of PAI-1 and visfatin: a randomized controlled study", the authors provide a thorough statistical analysis of the hormonal levels in blood and gastrointestinal symptoms of IBS patients before and after dietary intervention with SSRD. While the paper would definitely deserve to be considered for publication in the Special Issue "Gluten-Free Diet and Gastrointestinal Diseases", several issues need to be addressed to finalize it.

The Abstract ends too abruptly. I would recommend the authors to add one more sentence as a conclusion.

Reply: We have revised the abstract and added the other values that also differed between healthy volunteers and IBS patients. We have added a conclusion at the end of the abstract.

In Figures 1 and 2, some parts of the text do not fit the rectangles so they cannot be seen.

Reply: We have now changed so you can see all the text in both figures.

The diagrams in Figures 3, 4, and 5 are too small, especially labels on the OY axis. Also, units of measurement should be added.

Reply: Units are added in the figure descriptions. Further, we have improved the labeling of the axis with bigger figures.

Table 3 lacks the data on ghrelin, while table 5 lacks the list of abbreviations as in other tables.

Reply: The list of abbreviations is added in table 5. The intra- and inter assay coefficient of variance was not possible to calculate on ghrelin. This is explained on page 7, line 216-217, and the lines are deleted from Table 3. The low CVs of all analyses except ghrelin is a strength and limitation, respectively, and now mentioned on page 17, line 453-454.

Reviewer 2 Report

Main queries/suggestions for study team

  1. Is there sample size calculation done for the primary outcome of the study? What is the reason for unequal allocation of subjects to intervention (n=80) vs control (n=25) group?
  2. What are the recruitment criteria for the “healthy controls” in the secondary aim? Are they generally free of chronic illnesses (eg. Non-diabetic) or simply not diagnosed with IBS? Why are different number of controls used for different parameters?
  3. Critical study information such as the time of blood draw, whether participants fasted or consume anything before blood draw need to be included.

Line 28-29: It is not clear “controls” in this case refer to the ordinary diet group or healthy volunteers

Line 48 –51: It is not clear what the authors are trying to convey in this paragraph. If the effect of AXIN1 is confounded by endometriosis, did the team screened participants for endometriosis?

Line 204: How is intra-assay and inter-assay CV determine? What does the different N number mean? What is the significance of including this in the manuscript?

Line 98-125: Figure caption should be below the figures. Suggest combining Figure 1 and 2. There are repeat of information.

Line 114-117: What is the reason for unequal allocation of subjects to intervention (n=80) vs control (n=25) group?

Line 120-125: It is reported that 7 subjects were lost to follow up, but still all 105 subjects are used in the analysis. Please clarify.

Line 202 –265: Excessive information on ELISA kit and its experimental methods. Suggest shortening and condensing the useful information.

Line 295: Weight decreased after the intervention, could be a significant confounder for the results obtained for intervention group.

Line 317: It is not clear what the authors meant by responders? Responders of the survey or of the intervention? The percentage reported does not add up to 100%. What does the remaining % consist of? How come control group also has “responders” (of intervention).

Line 392 -393: Authors should be more careful with the word “controls”. Is this healthy “control” or control group of your intervention? In this case, the comparison seemed to be with the secondary aim, which is healthy participants?

Line 402: The changes/reduction was not significant at p=0.066.

Line 417: Postprandial? How long after a meal?

Line 493: When is the blood samples being obtained, if not fasted. Gut hormones can vary drastically under non-fasted condition.

Author Response

Dear Reviewer.

Thank you very much for reviewing our manuscript: Manuscript ID: nutrients-1636282;
Title: A starch- and sucrose-reduced diet in irritable bowel syndrome leads
to lower circulating levels of PAI-1 and visfatin: a randomized controlled
study

We have now tried to answer all the questions and think that the manuscript has been improved after revision. All changes are marked in yellow. We have also performed a language correction, but all these small changes throughout the manuscript are not marked. A detailed method description of ELISA and mesoscale is added as supplementary material, since one of the reviewers wanted a shorter method description within the manuscript. We now hope that you will further consider the manuscript for publication in Nutrients.

Bodil Ohlsson, professor, senior consultant in Gastroenterology and Hepatology

Reviewer 2.

Main queries/suggestions for study team

1.Is there sample size calculation done for the primary outcome of the study? What is the reason for unequal allocation of subjects to intervention (n=80) vs control (n=25) group?

Reply: We agree that the sample size is unequal. The initial aim of the study was to analyze the SI genetics and study the effect of the diet in relation to the functional SI genes. Therefore, we wanted as many participants as possible in the intervention group, to get a high number of participants. We still wanted also a control group to be able to compare the intervention and control diet. This is now added and explained as a limitation of the study, page 17, line 457-459.

2.What are the recruitment criteria for the “healthy controls” in the secondary aim? Are they generally free of chronic illnesses (eg. Non-diabetic) or simply not diagnosed with IBS? Why are different number of controls used for different parameters?

Reply: All healthy controls had to complete the study questionnaire and VAS-IBS as the patients, page 7. Line 202-203. Thus, we have been able to exclude any chronic diseases or gastrointestinal symptoms in controls. The reason for different numbers depends on that we first analyzed AXIN1. In the meantime, we collected and included further controls at the laboratory, as we then could use for the hormonal analyses. We could not complete with further control samples of AXIN1, since all analyses had to be performed at the same time and with the same batches.

3.Critical study information such as the time of blood draw, whether participants fasted or consume anything before blood draw need to be included.

Reply: The blood samples were not collected at fasting, but at the same time points at

both occasions within the same patient, page 5, line 154-155. We also performed within-group calculations, to compare the participants with themselves, to reduce the risk of non-fasting blood samples. Since the adipokines are not dependent on food intake, this should not affect the values of PAI-1 and visfatin (Al-Suhaimi et al. Eur J Med Res 2013). However, we have added this as a limitation of the study, since it may affect some of the other hormonal levels, page 17, line 454-457.

Line 28-29: It is not clear “controls” in this case refer to the ordinary diet group or healthy volunteers

Reply: This means healthy controls, which has been clarified now. We have changed healthy controls to healthy volunteers throughout the whole manuscript, to make it easier to differ between these controls and the control group in the intervention.

Line 48 –51: It is not clear what the authors are trying to convey in this paragraph. If the effect of AXIN1 is confounded by endometriosis, did the team screened participants for endometriosis?

Reply: All IBS patients had to report all kinds of medicines and medical health history. Only one patient had endometriosis and 3 used birth control medicines, which is the treatment of choice for endometriosis, page 9, line 282-283. But we have not performed ultra-sound or laparoscopy to exclude endometriosis. We mean that we must estimate the levels of AXIN1 in an IBS cohort, before we can use AXIN1 to differ between AXIN1 and IBS. This is now better explained in the introduction, page 2, line 52-53 and in the discussion, page 17, line 446-448.

Line 204: How is intra-assay and inter-assay CV determine? What does the different N number mean? What is the significance of including this in the manuscript?

Reply: The coefficient of variation is the standard deviation divided by the mean and expressed by percentage (CV%: (Standard deviation/Mean)*100). CV is used in analytical chemistry to express the precision and repeatability of an assay. Inter-assay CV = between different plates on different days, Intra assay CV = Same day on same plate. N = number of values, the higher number, the safer value. CV<10 is very good, 10-20 is good, 20-30 is acceptable and >30 is not acceptable.

Line 98-125: Figure caption should be below the figures. Suggest combining Figure 1 and 2. There are repeat of information.

Reply: Figure captions are now moved below the figures. We do not think we can combine these two figures, since figure 2 is the CONSORT flow chart, and the other is a much more detailed flow chart to explain the whole study design and inclusion and exclusion criteria. It gives a better overview than to write it in the text section.

Line 114-117: What is the reason for unequal allocation of subjects to intervention (n=80) vs control (n=25) group?

Reply: We agree that the sample size is unequal. The initial aim of the study was to analyze the SI genetics and study the effect of the diet in relation to the functional SI genes (Henström et al. Gut 2018). Therefore, we wanted as many participants as possible in the intervention group, to get a high number of participants. We still wanted also a control group to be able to compare the intervention and control diet. This is now added and explained as a limitation of the study, page 17, line 454-457.

Line 120-125: It is reported that 7 subjects were lost to follow up, but still all 105 subjects are used in the analysis. Please clarify.

Reply: Since we performed calculations between IBS patients and healthy volunteers at baseline, we could use all 105 patients at baseline, although some of them were lost at follow-up and could not be used in the pair-wise calculations.

Line 202 –265: Excessive information on ELISA kit and its experimental methods. Suggest shortening and condensing the useful information.

Reply: We have now shortened the method description, but added the whole initial description as a supplementary material since we think this is valuable for the reader if they want to repeat data.

Line 295: Weight decreased after the intervention, could be a significant confounder for the results obtained for intervention group.

Reply: Yes, this is correct. We have performed correlations between changes in hormonal levels and changes in symptoms, nutrient intakes and weight. Very few correlations were found, and all significant correlations are shown on page 15, line 366-373. The weight changes only correlated with CCK changes. Neither carbohydrates, starch, or sucrose correlated with hormonal levels, and these are the nutrients most interesting and affected by the diet (Nilholm et al. Nutrients 2021). Independently on which nutrient that led to the decreased values, sucrose or starch, this diet renders weight reduction. And it may be the combination of different changes in food intake that is of importance rather than a single item. Weight reduction per se is important for these hormonal levels. Thus, both direct and indirect effects of a diet are of importance for the health. Weight reduction does not seem to be a confounder since it did not correlate with changes in PAI-1 and visfatin levels. We also have performed Generalized linear model with hormonal changes as dependent value and nutrient intake as a predictor with weight as confounder. These calculations did not show any significance and are therefore not described in the manuscript.

Line 317: It is not clear what the authors meant by responders? Responders of the survey or of the intervention? The percentage reported does not add up to 100%. What does the remaining % consist of? How come control group also has “responders” (of intervention).

Reply: Responders are defined by a decrease in total IBS-SSS of ≥ 50 points (Francis & Morrsi APT 1997). This is now explained, page 11, line 304.

We mean that 73% were responders, and thus, 27% were not responders. 18% of the intervention group were completely out of symptoms. Thus, some of the responders were cured. Since the IBS has a fluctuating pattern, symptoms may aggravate or improve during time. This can explain the improvement in some participants of the control group. Further, there may be a placebo effect just by taking part in a clinical trial, and some of the participants in the control group improved their dietary habits during the study, since they noticed on their own in the diary protocols that they had poor food habits. The poor food habits are reported previously in Nilholm et al. Nutrients 2019.

Line 392 -393: Authors should be more careful with the word “controls”. Is this healthy “control” or control group of your intervention? In this case, the comparison seemed to be with the secondary aim, which is healthy participants?

Reply: We have now gone through the manuscript and tried to be more exact in the description of control group of diet and healthy controls, calling the latter healthy volunteers to make it easier.

Line 402: The changes/reduction was not significant at p=0.066.

Reply: You are right. Therefore, we throughout the manuscript call it a tendency or tended to be significant.

Line 417: Postprandial? How long after a meal?

Reply: Blood samples were non-fasting, as explained on page 5, line 154-155 and mentioned as a limitation on page 17, line 454-457. The blood samples were mostly collected 1-2 hours after a meal. The participants came to the scheduled time point at the same time point at both occasions, and participants were compared with themselves before and after the intervention. The fat intake was the intake during a whole day.  

Line 493: When is the blood samples being obtained, if not fasted. Gut hormones can vary drastically under non-fasted condition.

Reply: The blood samples were not collected at fasting, but at the same time points at both occasions within the same patient, page 5, line 154-155. We also performed within-group calculations, to compare the participants with themselves, to reduce the risk of non- fasting blood samples. You are right that gut hormone levels may vary drastically depending on food intake.  Therefore, we collected blood samples at the same time point on both occasions. Since the adipokines are not dependent on food intake (Al-Suhaimi et al. Eur J Med Res 2013), this should not affect the values; and these values are the values of interest in the present study. However, we have added this as a limitation of the study, since it may affect some of the hormonal levels, page 17, line 454-457.

References

Al-Suhaimi, E.A.; Shehzad, A. Leptin, resistin, and visfatin: the missing link between endocrine metabolic disorders and autoimmunity. Eur J Med Res 2013,18,12.

Francis, C.Y.; Morris, J.; Whorwell, P.J. The irritable bowel severity scoring system: a

Simple method of monitoring irritable bowel syndrome and its progress. Aliment Pharmacol

Ther 1997,11,395–402.

Henström, M.; Diekmann, L.; Hadizadeh, F.; Zheng, F.B.T.; Assadi, G.; Kuech, E.M.; Dierks,

C.; Heine, M.; Philipp, U.; Distl, O. et al. Functional variants in the sucrase-isomaltase gene

associate with increased risk of irritable bowel syndrome. Gut 2018,67,263-270.

Nilholm, C.; Larsson, E.; Roth, B.; Gustafsson, R.; Ohlsson, B. Irregular Dietary Habits

with a High Intake of Cereals and Sweets Are Associated with More Severe Gastrointestinal

Symptoms in IBS Patients. Nutrients 2019,11,1279.

Nilholm, C.; Larsson, E.; Sonestedt, E.; Roth, B.; Ohlsson, B. Assessment of a 4-Week Starch- and Sucrose-Reduced Diet and Its Effects on Gastrointestinal Symptoms and Inflammatory Parameters among Patients with Irritable Bowel Syndrome. Nutrients 2021,13,416.

Round 2

Reviewer 2 Report

The authors have addressed issues that can be addressed in its present stage However, fundamentally, this study is not well designed and has numerous limitations which the authors have also acknowledged.